# A 3D map of englacial attenuation rate from radar reflections at Law Dome, East Antarctica

Syed Abdul Salam<sup>1</sup>, Jason L. Roberts<sup>2,3</sup>, Felicity S. McCormack<sup>1,4</sup>, Richard Coleman<sup>1</sup>, and Jacqueline A. Halpin<sup>1</sup>

<sup>1</sup>Institute for Marine and Antarctic Studies, University of Tasmania, Hobart, Tasmania, Australia

<sup>2</sup>Australian Antarctic Division, Hobart, Tasmania, Australia

<sup>3</sup>Antarctic Climate & Ecosystems Cooperative Research Centre, University of Tasmania, Hobart, Tasmania, Australia <sup>4</sup>School of Earth, Atmosphere & Environment, Monash University, Clayton, Victoria, Australia

Correspondence: Syed.AbdulSalam@utas.edu.au

**Abstract.** The East Antarctic Ice Sheet (EAIS) is the largest source of potential sea-level rise, containing approximately 52 m of sea level equivalent. To constrain estimates of sea level rise into the future requires knowledge of ice-sheet properties and geometry and ice penetrating radar offers a means to estimate these properties (e.g. ice thickness, englacial temperatures). One of the regions that has been extensively surveyed using ice penetrating radar from the Investigating the Cryospheric Evolution

- of the Central Antarctic Plate (ICECAP) project in East Antarctica is Law Dome, a small independent ice cap situated to the west of Totten Ice Shelf. The ice cap is slow-moving, has a low melt-rate at the surface and moderate wind speeds, making it a useful study site for estimating the radar attenuation. A new method is proposed for the estimation of attenuation rate from radar data which is mathematically modelled as a constrained regularised  $l_2$  minimisation problem. In the proposed method, only radar data is required and the englacial reflectors are automatically detected from the radar data itself. To validate our
- methodology, attenuation differences at flight crossover points are calculated and statistical analyses performed to assess the accuracy of the results. For spatial analyses, the errors are of the order 22.6%, 15.2%, and 32.8% for mean absolute error, median absolute error, and root mean square error respectively. Also, for the depth analyses, up to the depth of 800m, the errors are under 29.9%, 24.2%, and 38.8% for mean absolute error, median absolute error, and root mean square error respectively. A final product of 3D attenuation rates and uncertainty estimates is provided. The generated dataset is publicly available at
- https://doi.org/10.25959/5e6851e266f4a (Abdul Salam, 2020).

# 1 Introduction

The East Antarctic Ice Sheet (EAIS) is the world's largest source of sea-level rise, with the marine-based component (where the ice-sheet is grounded below sea level) containing enough ice to raise sea levels by  $\sim$ 52 m (Morlighem et al., 2020). The rate of the potential EAIS contribution to sea-level rise can be estimated by calculating the ice-sheet mass budget. Ice dynamics

play an important role in the ice-sheet mass budget by transporting ice from areas of accumulation to areas of mass loss. Ice transport is strongly influenced by the internal temperature distribution within the ice-sheet, with warmer, more deformable ice leading to potential increases in ice-flow velocities (Easterbrook, 1999; Greve and Blatter, 2009), greater discharge of ice

into the ocean, and global sea-level fluctuations. Estimating attenuation rates and temperatures from ice penetrating radar has the benefit of constraining important model parameters (e.g. GHF in the thermal model) that can be used in simulations of present-day ice dynamics, as well as simulations of AIS contribution to future sea level rise. 25

Radar-echo sounding (RES) is a powerful and widely used geophysical method to characterise the physical properties of the ice medium, such as ice thickness, temperature, and englacial structure of ice-sheets and glaciers (Gudmandsen, 1971; Peters et al., 2005; Bingham and Siegert, 2007; MacGregor et al., 2007, 2012, 2015; Bogorodsky et al., 2012; Schroeder et al., 2015). The basis for RES is the detection of transmitted electromagnetic waves reflected from within the ice-sheet; the attenuation,

transmission and reflection of these waves is controlled by the electrical conductivity and permittivity of the ice (Reynolds, 30 2011). As conductivity and permittivity are functions of the ice chemistry, they vary spatially and through the ice column (or englacially), impacting the strength of the reflectivity, and enabling the exploration of the physical properties of englacial ice structures (Peters et al., 2005; Schroeder et al., 2016). The attenuation rate is primarily controlled by the ice-sheet's temperature and chemistry (Matsuoka et al., 2010, 2012; MacGregor et al., 2012), and can be used to characterise the physical properties 35 of ice.

Previously, many studies assumed englacial attenuation rates to be uniform within the ice and proportional to the ice thickness (Bentley et al., 1998; Rippin et al., 2004; Jacobel et al., 2010). However, the attenuation rate varies with location, due to changes in ice properties (Matsuoka et al., 2011; MacGregor et al., 2012). For example, a study in central West Antarctica shows that the one-way depth-averaged attenuation rate varies horizontally by 5  $dB \ km^{-1}$  in absolute terms along radar tran-

sects (Matsuoka et al., 2010). Matsuoka et al. (2011) argued that the assumption of regionally uniform attenuation rate fails 40 in most cases due to varying ice chemistry and temperature which leads to false attenuation estimates. However, these studies explored attenuation rates only in small regions and the understanding across broader spatial scales needs to be improved.

MacGregor et al. (2015) demonstrated the use of radar reflections from englacial layers to constrain the attenuation rates and temperatures as a function of depth in the Greenland ice-sheet. A key advantage of the method derived by MacGregor et al.

- (2015) is that it does not rely on echoes from the ice-sheet bed to determine attenuation, as the bed is complex and spatially 45 variable (Winebrenner et al., 2003), complicating interpretation of radar data near the bed. The disadvantage of this method is that it requires radiostratigraphy (study of layering by means of radar reflections) data to reliably trace the englacial layers (MacGregor et al., 2015). Many studies considered inferring the englacial attenuation or temperature from airborne radar data (Robin et al., 1969; Barrella et al., 2011; MacGregor et al., 2012, 2015; Schroeder et al., 2016), but none have done it in a way 50 that only relied on radar data.

Another radar-based method to constrain attenuation rates was proposed by MacGregor et al. (2007). In this method it assumed constant reflectivity values for the internal layers and the method also exploits the dependence of radar attenuation on ice temperature. However, this approach requires undisturbed englacial layers (i.e., having clear boundaries in englacial layers), which cannot be applied to many important regions because of the lack of englacial layer information. This depth-

averaged method is applicable at the ice-sheet scale, which requires contiguous and undisturbed englacial layers, and cannot be 55 applied to areas which include highly-crevassed and fast-flowing regions near grounding zones and shear margins (MacGregor

et al., 2015). However, as these regions often influence the dynamics of the ice-sheet to a greater extent layer-based approaches cannot be used to obtain englacial temperatures in many critical areas for ice-sheet modelling.

- As an alternative to the methods described above, Schroeder et al. (2016) developed an adaptive approach for estimating englacial attenuation rates for the entire ice column in Thwaites Glacier, West Antarctica. In this method, the unfocused radar bed echoes are fitted based on the correlation of ice thickness and the corrected bed power echo. The method performs weakly across the catchment near steeply sloping bed topography. In addition, another concern is that this method returns a 2D spatial map; however, deriving englacial temperatures requires estimates of attenuation throughout the ice column, requiring a 3D attenuation map. The temperature gradient can then be used to constrain the geothermal heat flux in the ice-sheet.
- The key motivation for this work is to enable the mapping to englacial temperature profiles possible, which can help in estimation of geothermal heat flux. Accurate estimates of englacial temperature and geothermal heat flux are incredibly important for constraining model simulations of ice dynamics (e.g. viscosity is temperature-dependent) and sliding. However, we currently have few direct measurements of vertical temperature (i.e. only at boreholes/ice domes) and geothermal heat flux in Antarctica. This method derives attenuation rates, that can then be mapped directly to englacial temperatures and geothermal
- heat flux. Several earlier studies have derived Arrhenius based functions to relate attenuation and ice temperatures in Greenland (MacGregor et al., 2015; Jordan et al., 2016) and West Antarctica (MacGregor et al., 2007). A mapping function can also be derived using a temperature profile as output and attenuation rates (for a certain radius around the borehole) as input. A temperature profile from the ice-borehole at Dome Summit South is available (Dahl-Jensen et al., 1999) which can help in generating mapping function. The attenuation rates derived here can then be transformed to temperature profiles by utilising
- the mapping function. Geothermal heat flux can be estimated from the gradient of these vertical temperature profiles (working paper).

Our proposed method overcomes some of the limitations of the previous methods. Initially, the englacial reflectors are detected in the reflection data and then the mathematical technique that minimises the mismatch is used to obtain the estimates of englacial attenuation. This method does not require any additional data-sets (such as englacial stratigraphy). Therefore, this

- method can be applied to regions which only have radar reflection data. A large amount of radar data has already been acquired for a number of purposes across vast areas of the ice-sheet. In East Antarctica, stratigraphy is not available except for limited regions typically around ice coring sites (e.g. at Dome C, where some internal layers are traced (Cavitte et al., 2018)), and while initiatives such as AntAchitecture aim to address this issue in the longer term, the majority of radar data is currently without stratigraphy. This method can utilise the existing acquired data and will extract the attenuation rates, which can further
- be used to estimate temperature profiles; consequently, it can improve the geothermal heat flux maps and ultimately can help improve ice-sheet modelling predictions. The 3D map of englacial attenuation rate is not available from any previous study and it can be a good data set for testing different novel algorithms. Here, the radar data from the Investigating the Cryospheric Evolution of the Central Antarctic Plate (ICECAP) (Young et al., 2011) mission for Law Dome East Antarctica are used. In the next section, the importance of the Law Dome region is discussed. Section 3 describes the methodology used for estimating
- the englacial attenuation rates. An assessment of the quality of the results, crossover and uncertainty analyses are presented in

Section 4. A summary of the research findings is presented in section 5, including the derived 3D map of attenuation rates and their uncertainty.

### 2 Study region and input data

Law Dome (66.7°S, 112.8°E) is a small independent ice cap (approximately 200 km in diameter) which is situated on the coast
of East Antarctica as shown in Fig. 1. Law Dome is a suitable case study for estimating attenuation for a number of reasons: 1)
the availability of well-sampled radar data from the ICECAP (Young et al., 2011) project, an aerogeophysical survey over East
Antarctica; 2) the ice core drilling site of an approximately 1200 m deep surface-to-bedrock at Dome Summit South (DSS)
which is located approximately 4.7 km SSW of the Law Dome Summit at an elevation of 1370 m (Dahl-Jensen et al., 1999;
Roberts et al., 2015) which can be helpful in mapping attenuation to temperature profiles; and 3) the ice on Law Dome is
slow-moving and stable, and there is negligible melting at the ice surface (Etheridge et al., 1998) and relatively moderate wind speed (8.3 ms<sup>-1</sup>) (Morgan et al., 1997), which means that the englacial layers are relatively undisturbed.

The ICECAP geophysical data that we use in this study were collected over the period 2008-2012, covering over 14800 linekm of the Law Dome region (Blankenship et al., 2011; Young et al., 2011). The survey aircraft was fitted with a High Capability Radar Sounder (HiCARS) instrument, with a central frequency of 60 MHz and bandwidth of 15 MHz. In order to retain full

- energy of the radar reflections, the radar data was processed after pulse compression (Wright et al., 2012). Young et al. (2011) determined the ice thickness from ICECAP airborne radar data using an ice speed of 169  $m/\mu s$  for electromagnetic wave propagation and no firn correction was applied for the reduced velocity of radio-waves. In this work the bed-echo and aircraft height data, ice thickness, and radar reflections will be used in order to extract the attenuation rate. Figure 1 shows the coverage of these data sets from ICECAP project for the entire Law Dome region.
- The spatial resolution of the ICECAP radar data is approximately 20 m along the flight line while the spacing between the flight lines is in order of kilometres and is not evenly sampled as shown in Fig. 1.

# 3 Methodology

We use radar reflectors from within the ice column to estimate radar attenuation. Radar attenuation is the loss (dB) of signal strength due to dielectric absorption, scattering and geometrical spreading. The attenuation rate given by following Equation 1 describes the loss of energy from source to receiver per unit distance (Reynolds, 2011).

$$N = \omega \left\{ \frac{\epsilon}{2} \left[ \left( 1 + \frac{\sigma}{\omega^2 \epsilon^2} \right)^{1/2} - 1 \right] \right\}^{1/2} \tag{1}$$

The attenuation rate (N) strongly depends on electrical conductivity  $(\sigma)$ , permittivity  $(\epsilon)$  and angular frequency  $(\omega)$ . As some of these properties are themselves strongly dependent on temperature, variations in the attenuation coefficient can be used to estimate englacial temperature, subject to making assumptions, such as constant conductivity (same chemical properties). Radar attenuation is proportional to conductivity and depends on the ice chemistry/acidity but it varies slowly spatially. In the

120 Radar attenuation is proportional to conductivity and depends on the ice chemistry/acidity but it varies slowly spatially. I case of Law Dome, the concentrations of impurities in the ice column are low (Etheridge et al., 1998).