# Peer review of "A 3D map of englacial attenuation rate from radar reflections at Law Dome, East Antarctica"

_Earth System Science Data, 2020_

## Referee Comment (RC1) · Anonymous Referee #1 · 9 Nov 2020

Review of "A 3D map of englacial attenuation rate from radar reflections at Law Dome, East Antarctica" by S. Abdul Salam et al.

8 November 2020

This manuscript describes the development and application of a new method for measuring englacial radar attenuation using an extensive radar dataset from Law Dome on the East Antarctic coast. The rationale for a new method is explained and multiple parameter decisions are considered in the new algorithm, which possesses the advantage of not requiring direct tracing of internal radar reflections. The resulting dataset

Explored within this manuscript is an intriguing idea that I first considered upon reading Hills et al. (2020, doi:10.1017/aog.2020.32): automatically classifying the "scope"

vertical traces of a radargram in terms of internal reflections to quickly trace prominent reflections and then derive attenuation rates. This manuscript aggressively pursues this idea using an appropriate dataset (they're forgiven for not knowing Hills et al. first presented part of it). However, in my view the manuscript overreaches and in a way that is beyond the scope of ESSD as I understand it. IMO, new methods should be previously validated for a journal focused on datasets, and also the presented datasets should have clear potential for wider use. I cannot presently assume that is likely of this attenuation-rate dataset, as I am still trying to fully unpack the method and the algorithmic uncertainties alone seem large. As such, I do not consider my review exhaustive as there are multiple major issues to be addressed and iterated on.

Major concerns

1. It's unclear whether the returned power peaks the algorithm detects are credibly interpreted as internal reflections rather than instrument or environmental noise. As best as I can tell, there is no assessment of the horizontal continuity of the candidate reflections between traces, which seems essential to making that case. One might instead argue that the present approach is a parsimonious way forward, but that specific argument is not made and the reader is left to assume that continuity is not important in this case. Perhaps the cross-over errors on the resulting attenuation rates support the argument that the signals are real enough? But that's not a great validation. If the candidate reflections don't meet some minimum threshold of horizontal continuity (10 km?), then I think they should be dropped. The 60-m depth interval for reflections seems reasonable, but no clear physical basis is provided for it (e.g., multiple of range resolution of radar system, qualitative examination of radargrams).

2. After reading the manuscript twice, it's still unclear to me what the "3-D" attenuation-rate field is, and I'm concerned that it's simply the gridded interval attenuation rate between each candidate reflection. If so, that would appear to explain the large range in attenuation rates discussed and the need for the N>0 constraint, including very low values, because this approach would place unjustified confidence in the assumption

of vertically uniform reflectivity. Layer reflectivities are noisy, and the vertically uniform assumption is the only present path toward an estimate of the *depth-averaged* attenuation rate across multiple (more than two) reflections (e.g., Matsuoka et al., 2010; MacGregor et al. 2015). Interval attenuation rates should only be calculated if the layer reflectivities are much better constrained than they currently are, and limited efforts on this topic have not yet demonstrated calculating interval attenuation rates are sufficiently fruitful to apply at the scale considered here (Holschuh et al., 2016, 10.1002/2016jf003942; Hills et al., 2020).

3. Is there any relationship between the attenuation-rate field and ice flow across Law Dome? This is not discussed at all, and so it's hard to know if the many potential uses of this dataset discussed in the introduction are actually relevant if there is no evaluation of the final product's potential utility?

Minor concerns

Abstract. I assume that the wind speeds are mentioned because of the potential for unconformities, but lots of places on both ice sheets are windy and don't have unconformities, nor is their effect upon attenuation rates known. So is this really necessary to mention in the abstract given his unclear relevance at best? The attenuation rate values are never mentioned in the abstract, which is odd given its centrality to the study.

16: The EAIS volume is better described as *potential* sea-level rise.

22: Easterbrook (1999) is an odd general reference for the statements made and a more specific one should be used.

26/7: Has any study actually used attenuation/temperature from radar to constrain GHF directly? I don't believe there are.

36: Those studies assumed the *attenuation* was proportional to ice thickness (not rate), so that a depth-averaged attenuation rate can be estimated.

71: "Mapping function" is an unusual term. Is this an inverse model? I realize these may be effectively synonyms, but IMO "mapping function" does a lot of work here for a task (temperature from attenuation in 3-D) that has not yet been demonstrated.

101: What is this wind speed? The mean annual value? Without further specificity it's meaningless.

110: Are these data SAR focused?

116: Specify at least once that this is the "real part of the relatively permittivity". "Permittivity" as it stands is vague.

117: Only one of the properties listed (conductivity) is strongly temperature-dependent.

---

## Referee Comment (RC2) · Kenichi Matsuoka (Referee) · 27 Nov 2020

This manuscript submitted to ESSD presents a new attempt to develop a map of radar attenuation rate over Law Dome, East Antarctica. Law Dome is a well-known, bench mark ice core site for Holocene climate and adjacent to fast flowing, rapidly changing Totten Glacier. Therefore, evolution of this ice rise, particularly stability of the dome position and non-climate-origin elevation changes, is of strong interest. Past evolution of the ice rise and geothermal flux can be inferred from the englacial thermal field, which can be possibly constrained by radar attenuation rate. Having it said, this paper addresses an important topic that has a broad impact. However, the results currently presented here are not robust enough to be immediately usable by others as is. In short, the method is not adapted well for this purpose, and the validation using the ice core is not carried out. Therefore, I would not recommend publication of this work in ESSD at this moment. I hope my comments below are useful for the authors to advance this work and justify my recommendation.

1. The method used here is based on vertical gradient analysis carried out in West Antarctic Ice Sheet Divide region (Matsuoka et al., 2010 in JGR, the reference list shows a wrong one in a proceedings) and later applied to Greenland Ice Sheet (MacGregor et al.). Their key argument justifying this analysis and the use of the vertical gradient as a proxy of attenuation rate is that large surface mass balance in those regions make upper half to two thirds of the ice sheet nearly isothermal (ice temperature stays mostly the same from the surface to these depths). Ignoring the secondary effects of ice chemistry, radar attenuation rate in this isothermal layer is then uniform. Radar power returned from englacial reflectors is a function of this depth-uniform attenuation rate and reflectivity of individual reflectors. They further assumed that reflector depths and reflectivity magnitudes are both randomly distributed, without strong depth trends (and this point was examined using Siple Dome ice core for the WAIS Divide work). Under these assumptions, the vertical gradient of the radar returned power can be (statistically) a proxy of ice temperature of this isothermal layer. In many cases, englacial radar reflectors are not recorded near the bed. Nonetheless, it is important to keep this analysis to the reflectors in the expected isothermal layer, because, below this isothermal layer, ice temperature varies largely from the temperature of the isothermal layer (near surface temperature) to the basal temperature near the melting point. Returned power near the deepest end of the analysis window has a strong influence on the vertical gradient when the returned power is fitted. These assumptions need to be discussed under given conditions at Law Dome, and examined using the Law Dome ice core (e.g. reflectivity and reflector depths). Some modifications are probably necessary for specific glaciological conditions at Law Dome. To be more specific, I'd analyze following to improve this work:
    a. Check Law Dome temperature record to see whether there is an isothermal layer or not, and how thick it is. Here, "isothermal" is for radioglaciology, so temperature difference of a few degrees is not a big matter.
    b. Examine radar data to see how deep significant englacial radar reflectors are recorded. The deepest echo to be analyzed should have a significant signal-to-noise ratio, at least a few decibels. Fig. 2 shows that the returned power is flattened at -60 dB around 800 m. Is this the noise level of this system?
    c. Derive acidity and salinity effects on radar attenuation using the Law Dome ice core data. Are they insignificant (so that they can be ignored throughout the entire analysis) or are they significant so that they should be considered or discussed? I am not familiar with previous work at Law Dome, but surface pit studies can be used to see possible spatial patterns of chemical fallouts over the dome. Combining the core data at the summit and pit data over the dome, can you get an insight how much are

acidity/salinity effects significant? Probably it is hard to implement this step in the analytical recipe but it is the point need to be discussed.

d. Similarly derive radar reflectivity using the acidity records of the core. Here, I would simply see the depth differential of acidity (because temperature is uniform enough and layer thickness can be ignored for the first attempt). Do reflector depths and reflectivity magnitudes have strong relationships with the depth, or are they mostly randomly distributed? Reflector depths can be examined from radar data as well, but radar data are affected by many other factors. So, I'd look into the ice core data for reflector depths too.

e. Analysis a and b above define the depth window that you can analyze. Analysis c and d above justify the analytical method and interpretation of the vertical gradients as the attenuation rate. These careful assessments should be made to have rigorous results, but the current manuscript does not present this framework clearly.

2. As I mentioned above, this method works in the top half to two thirds of ice thickness in the most cases (but actual depth range depends on many factors). Even if radar data show reflectors at greater depths, ice temperature varies with depth near the bed so that the isothermal assumption is invalid and thus the analytical method can not be applied there. Therefore, the thermal structure derived in this way is not sensitive to geothermal flux and more likely linked to the past glaciological evolution of the ice rise. I am curious whether resultant thermal structure has a clear contrast between the two sides facing to Totten Glacier and to outer sea. Probably this analysis alone cannot make a strong conclusion, but this analysis can be combined with other analyses to figure out possible past evolution and current boundary conditions using radar data. If the revised work will be submitted to another journal (not as data paper), I'd consider following points:

a. Suppose the derived radar attenuation is correct, the bed returned power (after the correction of the radar attenuation in the isothermal layer) is dominated by bed reflectivity and thermal structure of the deepest ice. Both are related to bed conditions (thawed or frozen) and probably to geothermal flux. Is bed uniformly frozen and geothermal flux nearly uniform, or do you see a specific spatial pattern?

b. Are there Raymond Arches beneath the dome summit? If so, can anomalous thermal structure derived by this work and Raymond Arches be discussed together to have the insights of past divide migration or differential thickness changes?

3. Ice core data are vital to justify the analytical method but also to validate the results. I would clarify the vertical gradient (direct outcome of the analysis) and englacial attenuation rate very carefully. The former can be a proxy of the latter, but they are not necessarily identical. Vertical gradient can be dependent of the radar system, but the attenuation rate is independent (the manuscript says that attenuation is system dependent, which is wrong, Line200). You can validate this relationship between the vertical gradient derived from the radar data and attenuation rate using the Law Dome ice core. Using the recipe developed in MacGregor et al. (2007), one can estimate the attenuation rate at the core site. Probably the derived vertical gradient is not precisely equal to the core-derived attenuation. Can chemistry data potentially explain this difference (as a secondary effect on the attenuation or as a reflectivity distribution)? Can the ice core pin down a possible uncertainty range of the radar-derived attenuation rate proxy?

I acknowledge that radar data can be analyzed in many ways, and I am not trying to promote my own paper. The analytical recipe I explained in #1 is not the golden rule and the principle can be implemented in many ways. I don't recommend this paper for publication, not because this work

does not follow the recipe I presented above, but because assumptions and analytical methods are not well articulated. Also, Law Dome has rich datasets, which benefit this work in a more efficient way. At least it is a great asset to validate the derived attenuation rate.

I wish that my comments above are useful for the authors.

Reviewed by

Kenny Matsuoka
Norwegian Polar Institute

---

## Editor Comment (EC1) · Reinhard Drews (Editor) · 18 Mar 2021

This paper has received two constructive reviews providing a number of suggestions for improvements in further iterations. It has been decided that no response-to-reviews will be provided and the review process will be discontinued at ESSD.

– Reinhard Drews (on behald of ESSD)